# Structural basis for hemoglobin scavenging by CD163 reveals mechanism of ligand promiscuity

Richard X. Zhou[1,2], Matthew K. Higgins[1,2]*

**1** Department of Biochemistry, University of Oxford, Oxford, United Kingdom, **2** Kavli Institute for Nanoscience Discovery, University of Oxford, Oxford, United Kingdom

* matthew.higgins@bioch.ox.ac.uk

## Abstract

The scavenger receptor CD163 detoxifies free hemoglobin released on erythrocyte lysis to prevent oxidative damage. The best understood route for hemoglobin detoxification involves the formation of haptoglobin-hemoglobin complexes that bind CD163 and are internalized into macrophages, resulting in hemoglobin degradation. However, during conditions such as sickle cell anemia or malaria, haptoglobin is depleted. CD163 can then act as a lower-affinity receptor for free hemoglobin. Previous studies revealed that CD163 forms a multimeric "base," which presents "arms" that form a binding site for haptoglobin-hemoglobin. In this study, we use cryogenic electron microscopy to reveal how human CD163 binds hemoglobin tetramers in a process that, unlike haptoglobin-hemoglobin uptake, requires a full trimeric CD163 assembly to achieve sufficient binding. We reveal how flexibility at the calcium-mediated base, combined with a hinge between receptor domains 2 and 3, allows the arms to wrap around diverse ligands. This brings together multiple small binding surfaces from different domains to form cradles for different ligands. These adaptations allow the scavenger receptor to be promiscuous, protecting us from oxidative damage caused by hemoglobin release in various pathological conditions.

## Introduction

Hemoglobin (Hb), carried within our blood, plays an essential role in transporting oxygen between our lungs and tissues. Hb is generally held within erythrocytes. However, hemolysis occurs due to erythrocyte damage and increases as a result of genetic conditions such as sickle cell anemia [1] or infections such as malaria [2]. Released Hb is toxic, with the haem group engaging in oxidative reactions and causing free radical generation [3–5]. Removal of free Hb from serum is therefore an important step in its detoxification, with physiological importance.

The best understood method of Hb detoxification involves the serum protein, haptoglobin (Hp), and the scavenger receptor, CD163. Hp first interacts with released Hb αβ-dimers, through a high-affinity interaction involving the serine protease (SP)

**Data availability statement:** Cryo-EM maps are deposited in the Electron Microscopy Data Bank with accession codes EMD-56135, EMD-56136, EMD-56137, and EMD-56138. Atomic coordinates are deposited in the Protein Data Bank with accession code 9TQD. All other data is presented in S1 Data.

**Funding:** Your current Financial Disclosure states "This work was funded by the Wellcome Trust. MKH is a Wellcome Investigator (220797/Z/20/Z to MKH) and RXZ was funded by the graduate program in Cellular Structural Biology (218482/Z/19/Z to MKH), Magdalen College and the Clarendon Fund (to RXZ). The funders had no role in study design, data collection and analysis, decision to publish, or preparation of the manuscript.

**Competing interests:** The authors have declared that no competing interests exists.

**Abbreviations:** CTF, Contrast Transfer Function; CV, column volumes; Hb, hemoglobin; Hp, haptoglobin; SPR, surface plasmon resonance; SRCR, scavenger receptor cysteine-rich.

domain of Hp [6,7]. Hp also contains multimerisation domains, which differ between isoforms 1 and 2. This results in different multimeric forms, from a dumbbell-shaped dimer found in those homozygous for Hp isoform 1, to larger multimers for heterozygous individuals or those homozygous for isoform 2 [8–10]. Formation of HpHb buries reactive groups on Hb, reducing their ability to cause oxidative damage [6]. HpHb is the primary ligand for the scavenger receptor CD163, found on macrophages, allowing its internalization and degradation [11].

Recent structural studies have shown how CD163 recognizes HpHb [12–15]. CD163 is a class I scavenger receptor, with an ectodomain that consists of nine scavenger receptor cysteine-rich (SRCR) domains linked to a C-terminal transmembrane helix [16]. SRCR5–9 forms a compact 'base' which mediates calcium-dependent multimerisation, resulting in dimeric and trimeric assemblies [12–15]. SRCR1–4 emerge as 'arms' from this base. In the unliganded receptor, the arms can be flexible or can associate through arm–arm interactions [12,15]. In the presence of HpHb, a single head structure, consisting of an SP domain of Hp and a Hb αβ-dimer (HpSPHb) binds a single CD163 dimer or trimer [12–15]. In each case, the arms of CD163 come together to form a single HpSPHb binding site, allowing binding and internalization of different multimeric forms of HpHb. Arm–arm interactions use the same surfaces as arm-ligand interactions, most likely resulting in autoinhibition of the receptor, thereby reducing receptor promiscuity by setting a threshold required for a ligand to compete with the arm–arm interactions to bind [12].

Structural and biophysical studies have also revealed the mechanism underlying HpHb internalization and release [12–15]. Mutant monomeric receptors mediate less efficient HpHb uptake [12]. Reductions in calcium concentration or pH also cause monomerisation as multimer formation is mediated by calcium ions bound to charged residues in the CD163 base [12–14]. In the absence of calcium or at low pH, the monomeric receptor also does not bind HpHb, as calcium ions mediate binding interfaces. This leads to a model in which cell surface CD163 multimerises through calcium binding and binds tightly to HpHb. However, when internalized, the lower calcium concentration and pH in the endosome results in CD163 monomerisation and dissociation of the ligand to be degraded within the macrophage.

While the primary ligand of CD163 is HpHb, increasing evidence suggests CD163-dependent internalization of Hb [12,13,17]. Hp is not required for Hb clearance in vivo as a Hp knock-out mouse can still clear Hb, albeit more slowly than wildtype mice [18]. Human polymorphisms (Hp0) which result in lack of Hp are also tolerated [19–21]. Indeed, CD163 has been shown to bind to Hb, although with affinities nearly 100-fold lower than those for HpHb [12,13,17]. Hb can also directly complete with uptake of labeled HpHb into cells, suggesting that Hb and HpHb bind to the same site on CD163, although with competition requiring higher concentrations of Hb than those of HpHb [17]. Also, uptake of Hb into human macrophages was significantly enhanced by the presence of Hp at low Hb concentrations, but not high concentrations of Hb [17].

These data indicate that small amounts of released Hb are internalized into macrophages using CD163 and a Hp-dependent route. However, Hp is present in human

sera at limited concentrations of 0.45–3.0 mg/ml (equivalent to 4.5–30 µM of Hp isoform 1 dimers) [22]. When hemolysis is high, due to infection or disease, Hp is depleted to undetectable levels [23,24]. In these conditions, Hb levels increase, for example, from 0.2 µM free heme in healthy individuals to 4.2 µM in those suffering from sickle cell anemia [23], and the Hb concentration exceeds that of Hp [21]. In these cases, Hp-independent Hb uptake can occur. To understand how CD163 binds to Hb in these situations, we used cryogenic electron microscopy to reveal the structure of the CD163-Hb complex. Comparison with structures of CD163 bound to HpHb shows how the receptor mediates promiscuity, allowing binding to either HpHb or Hb and resulting in Hb detoxification in different physiological conditions.

## Results

### CD163 is a hemoglobin receptor

We first aimed to confirm whether CD163 acts as a Hb receptor. We coupled CD163 ectodomain to a surface plasmon resonance (SPR) chip, capturing it through a C-terminal biotin to ensure presentation which matched that on a cell surface. Over this surface, we flowed Hb purified from human blood. As a comparison, we analyzed HpSPHb which has a similar shape to tetrameric Hb [25], the dominant species of Hb at micromolar concentrations [26]. The data for HpSPHb could be fitted to a one-to-one binding model with a $K_D$ of 1.18 nM, while the data for Hb was analyzed by equilibrium fitting with a $K_D$ of 382 nM (Figs 1A and S1).

We next assessed whether both Hb and a complex of the SP domain of Hp bound to an Hb αβ-dimer (HpSPHb) could be internalized into HEK293 cells in a CD163-dependent manner. We used a cell line which constitutively expresses full-length CD163 and incubated it in the presence of different concentrations of fluorescent Hb or HpSPHb, compared to a baseline of HEK293 cells not expressing CD163. We used fluorescence-activated cell sorting (FACS) to determine the amount of ligand uptake. Both Hb and HpSPHb were internalized in a CD163-dependent manner, with an $EC_{50}$ of 925 nM for Hb and 73 nM for HpSPHb (Fig 1B). Therefore, we confirm that HpHb and Hb are ligands for CD163, albeit with an affinity difference of ~300-fold and a difference in uptake efficiency of over 10-fold.

### The structure of hemoglobin-bound CD163

We next determined the structure of Hb-bound CD163. We assembled a complex of CD163 ectodomain with Hb in the presence of 2.5 mM $Ca^{2+}$, to match the calcium concentration found in serum and to ensure calcium saturation of the receptor. The complex was purified by size-exclusion chromatography, and grids were prepared for cryogenic electron microscopy (S2 Fig). Data were collected on a Titan Krios and processed using CryoSPARC [27] (S2 and S3 Figs and S1 Table), resulting in one predominant three-dimensional class (Fig 1C). The CD163 base, CD163 arms of subunits A and B and the Hb tetramer were well resolved, while the arm of subunit C was less well resolved. A model was built by docking individual SRCR domain models into the density before refinement.

The structure consists of a trimer of CD163 receptors which form a binding site for a single Hb tetramer. As in the case of HpHb-bound CD163 [12], domains SRCR5–9 from each receptor come together to form a triangular base. SRCR1–4 from each subunit emerge from this base as arms. Each arm makes different interactions with Hb, combining to form a binding site in which the Hb tetramer sits, with arm–arm interactions also forming between arms of subunits A and B (Fig 1C and S2 Table).

### Hemoglobin is bound by trimeric CD163

Our previous cryo-EM structures revealed that CD163 can form dimers and trimers, which exist both in the presence and absence of HpHb. In contrast, in the presence of Hb, we observed trimeric CD163 complexes, but no dimeric complexes. We therefore used analytical ultracentrifugation to assess the stoichiometry of CD163 in solution, when bound to HpSPHb or Hb (Figs 2A and S4). Analysis of the c(s) distribution for the CD163-Hb mixture showed a prominent peak

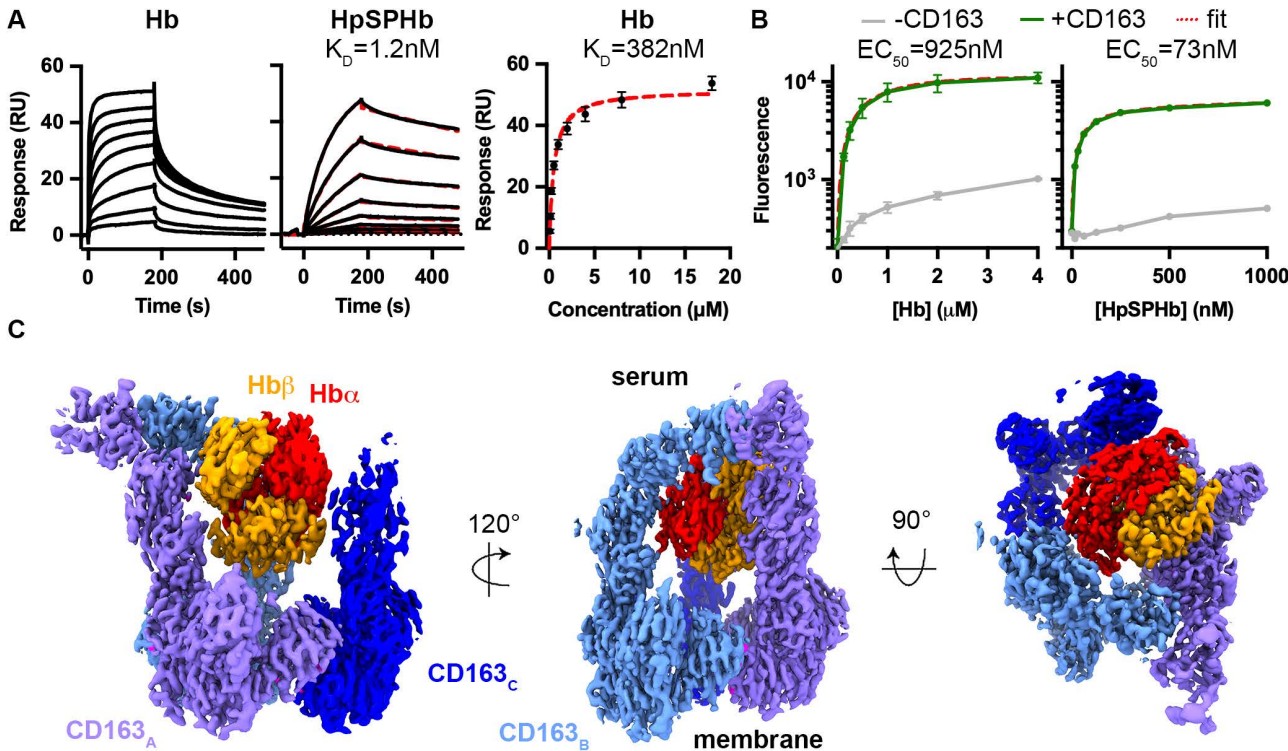

**Fig 1. Structural basis of hemoglobin binding by CD163. A.** Surface plasmon resonance analysis of the binding of Hb (left) or HpSPHb (central) to immobilized CD163. These represent 2-fold dilution series from 20 nM for HpSPHb and from 16 μM for Hb. The HpSPHb data could be fitted to a one-to-one binding model while the Hb data was fitted to an equilibrium binding model (right). Data representative of three repeats. **B.** Measurement of the uptake of fluorescently labeled Hb (left) and HpSPHb (right) into HEK293 cells expressing (green) or not expressing (gray) CD163. The red dashed line shows the fit. Data points represent the mean of three replicates, and error bars display standard deviations. **C.** The structure of a trimer of the CD163 ectodomain bound to a Hb tetramer. The three copies of CD163 are shown in three shades of blue, the α-subunit of Hb is red, and the β-subunit of Hb is orange and proposed calcium ions in magenta. The underlying data for Fig 1A and 1B can be found in S1 Data. The underlying data for Fig 1C is deposited in the Electron Microscopy Data Bank with accession codes EMD-56135, EMD-56136, EMD-56137 and EMD-56138 and the Protein Data Bank with accession code 9TQD.

at approximately 3.8 S, corresponding to free Hb tetramers, and another peak at approximately 13.6 S, representing the trimeric CD163-Hb complex. In contrast, the c(s) profile for the CD163-HpSPHb mixture revealed a more complex distribution with multiple peaks between 7 and 14 S. These peaks correspond to a mixture of species, including monomeric, dimeric, and trimeric CD163 bound to HpSPHb (Fig 2A). Therefore, while HpHb can bind CD163 monomers, dimers, or trimers, Hb was only found bound to trimers. This is most likely due to the higher affinity of CD163 for HpHb than for Hb, enabling both CD163 dimers and trimers to bind sufficiently tightly to HpHb to allow uptake, while trimers are required for Hb binding and uptake. Indeed, while Hb could compete for uptake of Hp(2-2)Hp into HEK293 cells transfected with CD163, it could not compete for uptake into cells transfected with a monomeric mutant of CD163, showing that monomeric CD163 cannot as effectively bind Hb as HpHb [12].

We next compared the architecture of the trimeric base of CD163 in the presence of either HpHb or Hb (Fig 2B). Our previous study revealed that interactions in the base are mediated by calcium ions, allowing pivoting of neighboring subunits around these ions. This allows rocking of the subunits and their arms relative to each other. We therefore compared the interactions within the base in trimeric CD163 bound to HpHb or to Hb. Alignment of SRCR5–9 from the Hb-bound and HpHb-bound CD163 trimers revealed a similar base architecture, with an overall RMSD of 1.9 Å, showing that

 

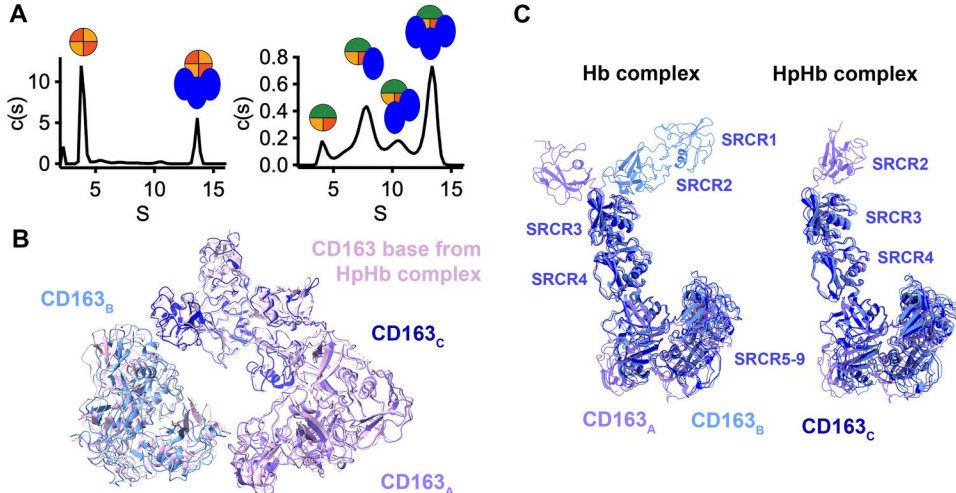

**Fig 2. Comparison of the architecture of Hb and HpHb-bound CD163. A.** Stoichiometry of complexes of 7 μM CD163 bound to 40 μM Hb (left) or 3 μM HpSPHb (right) as determined by analytical ultracentrifugation. Above the traces are symbols which indicate the likely identity of the species in the peak below the symbol, with the receptor as a blue oval, Hb α-chain in red, Hb β-chain in yellow and Hp in green. Shown are representatives of duplicate measurements. **B.** The base of trimeric CD163, consisting of domains SRCR5-9, from the Hb-bound complex (with the three CD163 monomers shown in different shades of blue) aligned to equivalent region of the trimeric CD163 base bound to HpHb (in pink). **C.** An alignment of the three CD163 monomers from the Hb-bound trimer (left) and the HpHb-bound trimer (right), aligned on SRCR5. In both cases, SRCR3-9 form a rigid structure, with a flexible hinge located between SRCR2 and 3 allowing distinct conformations of SRCR1 and 2. The underlying data for Fig 2A can be found in S1 Data. The underlying data for Fig 2B and 2C is deposited in the Electron Microscopy Data Bank with accession codes EMD-56135, EMD-56136, EMD-56137 and EMD-56138 and the Protein Data Bank with accession code 9TQD.

previously identified rocking around the subunit interfaces [12], is sufficient to switch from the Hb-binding to HpHb-binding conformations.

In the structure of CD163 trimers bound to HpHb, we found that the three CD163 subunits adopted a very similar conformation. To determine whether the same is the case in the Hb-bound conformation, we aligned the three CD163 subunits on SRCR5 (Fig 2C). Once again, we see that domains SRCR3–9 form a rigid unit. However, the interface between SRCR2 and 3 is more flexible, allowing SRCR1–2 to adopt different conformations from the rest of the subunit.

## Flexibility of the CD163 arms allows promiscuity

We next analyzed the nature of the ligand-binding site for Hb (Fig 3A). A single Hb tetramer binds to a site formed from all three arms of CD163, positioned slight off center relative to the CD163 trimeric base. The three CD163 arms all interact with different surfaces of Hb, with interactions mediated through small binding surfaces on the SRCR2–4 domains. Arm A interacts with both β-subunits of Hb, through the SRCR3 and 4 domains, with a sphere of density which we attribute to being a calcium ion mediating the former interaction (Fig 3B). Arm B engages one α-subunit of Hb, bound by SRCR2, 3, and 4. Arm C contacts an α-subunit of Hb through SRCR3 and a β-subunit through SRCR4. This accumulation of small binding sites collectively creates a cradle in which the Hb tetramer lies.

Comparison of the structures of the Hb- and HpHb-bound CD163 trimers shows how CD163 brings together a collection of small binding surface from individual SRCR domains to bind diverse ligands (Fig 3C). The same sites on SRCR3 and 4 can bind to either α- or β-subunits of Hb, depending on which arm and which ligand is bound. The same surfaces on SRCR2 can bind to either Hb or to Hp, completing the binding surface. The ability of the CD163 arms to tilt between SRCR2 and 3 allows them to wrap around their ligand, increasing the binding surface, and cradling a ligand of the size of Hb or HpSPHb.

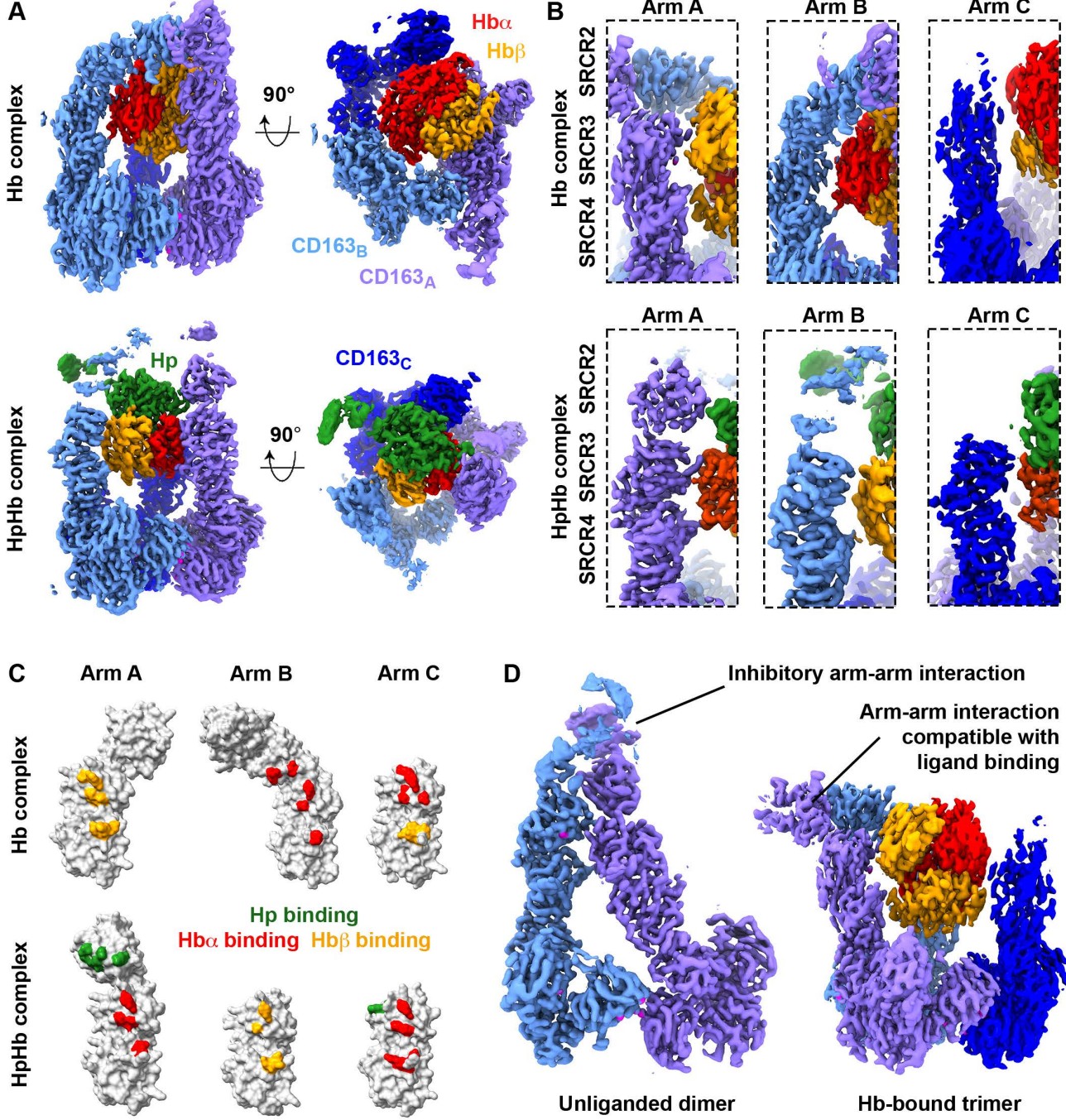

**Fig 3. Flexibility of the arms of CD163 allows binding to diverse ligands. A.** A comparison of the structure of trimeric CD163 bound to Hb (top) or HpHb (bottom). **B.** Close-up views of each arm of the CD163 trimer bound to Hb (top) and HpHb (bottom) showing the different arm-ligand interactions in each complex. **C.** A schematic showing the three arms of CD163 taken from the Hb-bound trimer (top) and HpHb-bound trimer (bottom), each represented as surfaces. In each case, residues binding to Hp are green, those binding to Hb α-chain are red, and those binding to the Hb β-chain are orange. **D.** Comparison of the structure of the unliganded CD163 dimer (left) and the Hb-bound trimer (right), showing arm–arm interactions which are incompatible with ligand binding in the unliganded, autoinhibited dimer, and those which are compatible with stabilizing the ligand-binding site in the Hb-bound complex. The underlying data is deposited in the Electron Microscopy Data Bank with accession codes EMD-56135, EMD-56136, EMD-56137 and EMD-56138 and the Protein Data Bank with accession code 9TQD.

Also notable is the diverse role of arm–arm interactions in the function of CD163. Our previous structure of unliganded CD163 showed that arm–arm interactions can occur through surfaces which are required for arm-ligand interactions [12] (Fig 3D). As these arm–arm interactions are not compatible with ligand binding, we proposed that this is an auto-inhibited conformation of CD163 and that these interactions reduce the binding of low-affinity ligands, as such a ligand would need to compete with arm–arm interactions to bind. In the structure of trimeric CD163 bound to Hb, we also see an arm–arm interaction between SRCR2 of arm A and SRCR1 of arm B (Fig 3D). In contrast to the unliganded state, these arm–arm interactions are compatible with Hb binding and may indeed stabilize the cradle in which Hb binds.

## Discussion

To perform its primary function as a scavenger receptor that mediates Hb detoxification, CD163 binds and internalizes a range of ligands. These include isoforms and multimeric states of HpHb found in different individuals. In addition, CD163 binds to free Hb tetramers, allowing it to contribute to Hb detoxification in conditions of Hp depletion. Structures of CD163 bound to HpHb and Hb tetramers show how it achieves these levels of ligand promiscuity, forming binding sites for similarly sized Hb tetramers and the HpSPHb head of different HpHb isoforms.

Promiscuity is possible as the architecture of multimeric CD163 allows it to bring together multiple small binding surfaces, each provided by a single SRCR domain, which combine to form a binding site. The surface areas used by each SRCR domain to bind to a ligand are small, each involving only a few bonds. However, multiple such interfaces combine, with seven SRCR domains each contacting Hb or HpHb, increasing affinity. To allow these different surfaces to reach the correct relative positions to bind to a ligand, two types of flexibility are observed. Firstly, tilting around the calcium-mediated interface joining SRCR5–7 moves the rigid SRCR3–9 units relative to each other, placing the lower part of the arms in the correct positions, as previously observed [12]. Secondly, flexibility in the interface between SRCR2 and 3 allows the SRCR1 and 2 unit to tilt as it wraps around the ligand, allowing further interactions between the ligand and SRCR2. This use of flexible arms, coming together in different conformations, generates cradles of different shapes for the two ligands. Finally, different degrees of multimerisation are possible, with trimerisation allowing uptake of the lower-affinity Hb ligand, while dimers and monomers are sufficient to bind and allow uptake of higher-affinity HpHb.

Arm–arm interactions also play varied roles in CD163. Arm-arm contacts which would compete with ligand binding occur in the unliganded receptor and are likely to set the affinity threshold below which arm–arm interactions are favored, reducing the degree of promiscuity. In contrast, other arm–arm interactions are compatible with binding and may even stabilize the formation of the ligand-binding pocket for a lower-affinity ligand. This synergy of three arms, with different degrees of flexibility, combined with multiple small binding surfaces and the different possible roles of arm–arm interactions, converges to determine the degree to which CD163 is promiscuous. These features can be used differently for the related ligands Hb and HpHb. Future studies will be required to show the depth of this promiscuity, for example, for other CD163 ligands, or its recognition by viral surface proteins.

## Methods

### Expression and purification of CD163

Human CD163 ectodomain (Uniprot Q86VB7-1, residues 46–1050) fused to a BAP-tag, GAA-linker, and C-tag was expressed in Expi293 cells (Thermo Scientific, A14635) in media supplemented with 0.1 mM biotin. Six days post-transfection, conditioned medium was clarified using a 0.45 µm filter and loaded onto CaptureSelect C-tagXL Affinity Matrix (Thermo Scientific). After washing in 30 column volumes (CV) of 20 mM Tris pH 7.5, 150 mM NaCl, CD163 was eluted in 5 CV of 20 mM Tris pH 7.5, 2 M $MgCl_2$. To increase protein yield, the flow-through was re-applied onto fresh C-tag resin. CD163 was polished using size-exclusion chromatography (Superose 6 Increase 10/300 GL column, Cytiva) into 20 mM HEPES pH 7.4, 150 mM NaCl, 2.5 mM $CaCl_2$, flash-frozen, and stored at −80 °C. In this manuscript, molar concentrations of CD163 are provided for a single CD163 subunit.

## Purification of Hb and production of HpSPHb

Hb was purified from fresh human red blood cell lysate using a Superdex 75 Increase 10/300 GL column (Cytiva) pre-equilibrated in 20 mM HEPES pH 7.4, 150 mM NaCl, 2.5 mM $CaCl_2$. HpSP (Uniprot P00738-2, residues 89–347) bearing a GAA-linker and C-tag was expressed and purified as CD163. HpSPHb was assembled by mixing purified HpSP with a molar excess of Hb and capturing the complex on C-tag resin. After washing away free Hb, HpSPHb was eluted and polished by SEC as above.

## Structure determination by cryo-EM

160 µg of the CD163:Hb complex were prepared by mixing CD163 trimer at 1.6 mg/ml with Hb tetramer at 2:1 molar ratio. 3 µl of the mixture was applied to glow-discharged Quantifoil R 1.2/1.3 Cu 300 grids in a Vitrobot Mark IV (Thermo Scientific) maintained at 4 °C and 100% humidity. Grids were blotted for 1.5–5 s and plunge-frozen into liquid ethane.

Data were acquired on a Titan Krios G3 (Thermo Scientific) operated at 300 kV, equipped with a K3 direct electron detector (Gatan) and a BioQuantum energy filter with a 20 eV slit (Gatan). Grids were imaged in EPU (Thermo Scientific) using faster acquisition mode, a magnification of 58,149×, a pixel size of 0.832 Å, a defocus range of −1.8 to −0.6 µm, and a total exposure of 2.4 s delivering 38.3 $e^-/Å^2$. A single grid was used for data collection and 23,892 movies were recorded.

## Image processing

All processing steps were performed in CryoSPARC v4.5 [27]. Movies underwent patch-based motion correction, Contrast Transfer Function (CTF) estimation, and template-based picking using a volume from a pilot dataset, yielding 11,892,554 particles. Particles were extracted in 512-pixel boxes and down-sampled to 384 pixels. Four rounds of 2D classification removed poorly defined classes, leaving 5,761,233 particles. These were subjected to ab initio reconstruction, followed by heterogeneous and non-uniform refinement, resulting in eight 3D classes. Two of these classes reached high resolution and showed high ligand occupancy.

Particles from the two classes were merged (2,647,984) for a second round of unsupervised 3D classification, leading to another eight classes. Two high-resolution classes with clear density for the $CD163_A$–$CD163_B$ arm–arm contact and nearly identical features were combined (1,033,649 particles), re-extracted at the original sampling (0.832 Å), refined to 2.8 Å following reference-based motion correction (denoted as map 1 in S2 Fig). The arm-arm interface was locally refined with a soft mask around SRCR1–2 of $CD163_B$ and SRCR2 of $CD163_A$ without particle subtraction, giving 3.1 Å (map 1, arm-arm contacts). A second class (477,310 particles) showed the $CD163_C$ arm at higher resolution; these particles were re-extracted, polished, and refined to 3.1 Å (map 2). A composite map was assembled in ChimeraX v1.7 [28] by combining the 2.8 Å consensus map with the $CD163_C$ arm and the $CD163_A$–$CD163_B$ arm–arm interface of the respective maps, then sharpened using DeepEMhancer [29] and visualized in ChimeraX.

## Model building and refinement

A single, composite model of trimeric CD163:Hb was built to fit the composite cryo-EM map. The triangular base of CD163 was placed by rigid-body fitting of the trimeric CD163:Hp(1–1)Hb base (PDB 9HEK) [12] into the corresponding density. SRCR domains forming the arms were positioned individually, with initial models for SRCR1 and SRCR2–4 obtained from AlphaFold 2 [30] and our previous CD163:HpHb structure [12], respectively. The $CD163_C$ arm and the $CD163_A$–$CD163_B$ arm–arm interface were modeled, as these regions were resolved in the composite map. For consistency with our earlier work, protomers were labeled $CD163_A$, $CD163_B$, and $CD163_C$ based on the inter-protomer spacing observed between SRCR7 of one subunit and SRCR9 of its neighbor. $CD163_A$ denotes the protomer whose SRCR9 engages SRCR7 of $CD163_B$ across the largest inter-protomer spacing at the base, and the third chain is labeled $CD163_C$.

Tetrameric, human oxygenated Hb (PDB: 1HHO) was docked into the ligand density. Model geometry was corrected in Coot 0.9.8.8 [31], clashes reduced with ISOLDE [32], and $Ca^{2+}$ ions and glycans were built where supported by map features. The structure was refined by iterative cycles of real-space refinement in PHENIX [33] and manual correction in Coot.

## Surface plasmon resonance

SPR measurements were carried out at 25 °C on a Biacore T200 instrument (Cytiva) with the Series S Biotin CAPture Kit (Cytiva) in SPR buffer (20 mM HEPES pH 7.4, 150 mM NaCl, 2.5 mM $CaCl_2$, 0.005% Tween-20). Before analysis, Hb, HpSPHb, and biotinylated CD163-WT were exchanged into SPR buffer using 0.5 mL Zeba spin columns (7 kDa MWCO, Thermo Scientific).

Because Hb binds only trimeric CD163, a high capture level was used to enrich trimers on the surface. Twelve µg/ml CD163-WT was injected at 8 µl/min for 160 s, giving ~570 RU. Hb was titrated in a 2-fold dilution series from 16 µM to 62.5 nM over the surface whereas HpSPHb was flown at 20 to 0.16 nM over the surface. Between cycles, the chip was regenerated with 6 M guanidine hydrochloride, 0.25 M NaOH for 80 s at 10 µl/min. Each experiment was repeated three times with three independent dilution series. Binding affinities of Hb to CD163 were calculated using steady-state fits to the response measured 4 s before the end of the association phase, using the "One site - Total" model in GraphPad Prism v10.6.0. For the HpSPHb-CD163 interaction, kinetic analysis was conducted using BIAevaluation software v1.0 (Cytiva) and a 1:1 binding model.

## Ligand uptake assay

Stably transfected, CD163-WT-expressing Flp-In-293 cells (Thermo Fisher cat: R78007) were generated as previously described [12]. Minimal labeling of lysines in the ligands is critical to prevent ligand inactivation. To further select for N-terminal labeling, 4 mg/ml Hb or HpSPHb were mixed with 0.15 mg/ml Alexa Fluor 594 NHS Ester (Thermo Scientific) in 50 mM phosphate buffer pH 7.4. After a 15 min incubation at room temperature, the reaction was quenched in 20 mM Tris pH 7.5, 140 mM NaCl, and labeled Hb or HpSPHb was separated from excess dye on a Superose 6 Increase 10/300 GL column (Cytiva) and buffer-exchanged into 20 mM HEPES pH 7.4, 150 mM NaCl, 2.5 mM $CaCl_2$.

For uptake, ~200,000 $CD163^+$ or untransfected cells were washed once with PBS and incubated for 30 min at 37 °C, 5% $CO_2$ in DMEM-HEPES (Thermo Fisher cat: 12430054) containing Alexa Fluor 594–labeled ligand (0–4 µM Hb or 0–1 µM HpSPHb). Subsequently, cells were washed with PBS, detached with trypsin, and washed again. DRAQ7 (Abcam) was added before acquisition to 3 µM to exclude non-viable cells. Fluorescence was recorded on an LSRFortessa X-20 flow cytometer (BD Biosciences) using the 488 nm laser for CD163-GFP and the 561 nm laser for the ligands.

Uptake assays were performed as three independent experiments on separate days, each using freshly prepared, ligand-containing medium and independently seeded cells. Data were analyzed in FlowJo v10.10 by calculating the mean population fluorescence of viable, single cells, and these values were visualized in GraphPad Prism.

## Analytical ultracentrifugation

To characterize the stoichiometry of the CD163-Hb and CD163-HpSPHb complexes in solution, sedimentation velocity experiments were performed. Samples were prepared by directly mixing 7 µM CD163 with either 40 µM Hb, 3 µM HpSPHb, or 40 µM HpSPHb in a buffer containing 20 mM HEPES pH 7.4, 150 mM NaCl, and 2.5 mM $CaCl_2$. These mixtures, with a final volume of 400 µL, were incubated for 2 h at room temperature before the run to allow for complex formation.

Data collection was conducted at 20 °C using a Beckman An-60 Ti rotor in a Beckman Optima XL-1 analytical ultra-centrifuge. For the 40 µM Hb and 40 µM HpSPHb mixture with CD163, sedimentation was monitored using interference

optics, whereas UV absorbance was utilized for the 3 µM HpSPHb sample with the receptor. The resulting scans were processed using SEDFIT [34] to generate continuous size-distribution c(s) profiles. Solvent parameters, including density and dynamic viscosity, along with the partial specific volumes of the protein components, were calculated from the amino acid sequences and buffer composition using SEDNTERP. The data, fit, residuals, and distribution curves were visualized using GraphPad Prism.

## Supporting information

**S1 Fig. Surface plasmon resonance data.** Surface plasmon resonance traces for binding of Hb (top) and HpSPHb (bottom) to immobilized CD163, each shown in triplicate. In the case of Hb, a 2-fold dilution series was used from a top concentration of 16 µM and data was fitted using equilibrium fitting. In the case of HpSPHb, a 2-fold dilution series was used from a top concentration of 20 nM. Data was fitted to a 1-to-1 binding model (red dashed lines), and the binding parameters are shown below the curves. The underlying data can be found in S1 Data.
(TIF)

**S2 Fig. Cryogenic electron microscopy processing scheme.**
(TIF)

**S3 Fig. Analysis of cryogenic electron microscopy data.** Gold-Standard Fourier Shell Correlation (GSFSC) plots for resolution estimate (top), maps showing local resolution (middle) and particle-view distribution (bottom) for map 1 (left), locally refined map 1 (central) and map 2 (right).
(TIF)

**S4 Fig. Analytical ultracentrifugation datas.** Analytical ultracentrifugation data for 7 µM CD163 in complex with 40 µM Hb (left), 3 µM HpSPHb (center), or 40 µM HpSPHb (right). In each case, two independent runs were conducted (top and bottom). Each set of three panels shows raw data (top), residual after fitting (middle), and fitted distribution data (bottom). Approximate molecular weights are shown above their corresponding peaks. The underlying data can be found in S1 Data.
(TIF)

**S1 Table. Cryo-EM data collection, refinement, and validation statistics.**
(DOCX)

**S2 Table. Table of interactions.**
(DOCX)

**S1 Data. Data from surface plasmon resonance analysis (Figs 1A, 1B, and S1); analytical ultracentrifugation (Figs 2A and S4), and cryogenic electron microscopy (S3 Fig).**
(XLSX)

## Author contributions

**Conceptualization:** Richard X. Zhou, Matthew K. Higgins.

**Formal analysis:** Richard X. Zhou, Matthew K. Higgins.

**Funding acquisition:** Matthew K. Higgins.

**Investigation:** Richard X. Zhou, Matthew K. Higgins.

**Methodology:** Matthew K. Higgins.

**Project administration:** Matthew K. Higgins.

**Supervision:** Matthew K. Higgins.

**Validation:** Richard X. Zhou, Matthew K. Higgins.

**Visualization:** Richard X. Zhou, Matthew K. Higgins.

**Writing – original draft:** Richard X. Zhou, Matthew K. Higgins.

**Writing – review & editing:** Richard X. Zhou, Matthew K. Higgins.

## Acknowledgments

The COSMIC facility provided the resources for cryo-EM data collection, and we thank Rishi Matadeen, Flavia Moreira-Leite and Ed Lowe. We acknowledge Hannah Ivison for lab management. Our gratitude goes to David Staunton for help with biophysical methods and to Robert Hedley and Vasiliki Tsioligka for assistance with flow sorting.

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
