## [Editor Report · Decision Letter 0]

6 Feb 2026

Dear Dr Higgins,

Thank you for submitting your manuscript entitled "Structural basis for haemoglobin scavenging by CD163 reveals mechanism of ligand promiscuity" for consideration as a Research Article by PLOS Biology. Please accept my sincere apologies for the delay in getting back to you with feedback.

Your manuscript has now been evaluated by the PLOS Biology editorial staff and I am writing to let you know that we would like to send your submission out for external peer review.

IMPORTANT: After discussions within the editorial team, we would like to consider your manuscript as a 'Short Report' at the journal (https://journals.plos.org/plosbiology/s/what-we-publish#loc-short-reports). Upon resubmission (see guidelines below), I would be grateful if you could tick 'Short Report' as the article type in the dropdown menu in the online submission form.

Before we can send your manuscript to reviewers, we need you to complete your submission by providing the metadata that is required for full assessment. To this end, please login to Editorial Manager where you will find the paper in the 'Submissions Needing Revisions' folder on your homepage. Please click 'Revise Submission' from the Action Links and complete all additional questions in the submission questionnaire.

Once your full submission is complete, your paper will undergo a series of checks in preparation for peer review. After your manuscript has passed the checks it will be sent out for review. To provide the metadata for your submission, please Login to Editorial Manager (https://www.editorialmanager.com/pbiology) within two working days, i.e. by Feb 08 2026 11:59PM.

Kind regards,

Richard

Richard Hodge, PhD

rhodge@plos.org

PLOS

---

## [Decision Letter · Decision Letter 1]

2 Apr 2026

Dear Matthew,

Thank you for your continued patience while your manuscript "Structural basis for haemoglobin scavenging by CD163 reveals mechanism of ligand promiscuity" was peer-reviewed at PLOS Biology. Please accept my sincere apologies for the delays that you have experienced during the peer review process. Your manuscript has now been evaluated by the PLOS Biology editors, an Academic Editor with relevant expertise, and by three independent reviewers.

In light of the reviews, which you will find at the end of this email, we would like to invite you to revise the work to thoroughly address the reviewers' reports.

As you can see, the reviewers are generally positive about your manuscript and think it is interesting and well done. However, the reviewers provide several concerns and experimental suggestions to strengthen the work. Reviewer #1 asks that the in vitro binding affinity and cellular uptake efficiency of monomeric/dimeric CD163 with Hb tetramer is tested. In addition, Reviewer #2 notes that the paper lacks essential cryo-EM validation data and requests that this information is provided so that the quality of the maps and model can be assessed. Finally, Reviewer #3 asks for a few clarifications and explanations, noting that they are confused by the stoichiometry of the CD163 base given that CD163 dimers are observed in samples prepared without substrate or with HpSPHb.

Given the extent of revision needed, we cannot make a decision about publication until we have seen the revised manuscript and your response to the reviewers' comments. Your revised manuscript is likely to be sent for further evaluation by all or a subset of the reviewers.

**IMPORTANT - SUBMITTING YOUR REVISION**

*Re-submission Checklist*

*Published Peer Review*

*PLOS Data Policy*

*Blot and Gel Data Policy*

Best regards,

Richard

Richard Hodge, PhD

rhodge@plos.org

REVIEWS:

Reviewer #1: The scavenger receptor CD163 clears free hemoglobin (Hb) during intravascular hemolysis thereby preventing oxidative damage. Typically, this process is effectively facilitated by the haptoglobin (Hp), which mediates a high-affinity complex formation between CD163 and Hb (termed CD163-HpHb). However, the tolerance observed in patients with genetic absence of Hp, or its conditional depletion during infection or sickle cell anaemia, suggests the existence of an Hp-independent, CD163-depedent clearance mechanism for Hb overload. In this study, the authors provide critical structural insights into this alternative CD163-Hb interaction. They demonstrate that the Hb tetramer engages exclusively with a trimeric form of CD163, in a pattern broadly similar to that of the HpHb ligand, despite a ~300-fold weaker affinity. A subtle difference lies in the local conformational rearrangement of ectodomains SRCR1/2, which may confer the flexibility needed for CD163 to recognize a variety of ligand configurations.

Overall, the manuscript is well-structured, methodologically rigorous, and addresses a biologically important question. The conclusions are supported by the data, and the comparative analysis with previous structural work strengthens the novelty of the findings. This study is suitable for publication in PLOS Biology as a Short Report, pending minor revisions to address the following points:

-A major finding of this study is the ligand-dependent selectivity in the oligomerization state of CD163. The cryo-EM and analytical ultracentrifugation data convincingly show that the Hb tetramer interacts only with CD163 trimer. Would it be feasible to further test the in vitro binding affinity and cellular uptake efficiency of monomeric or dimeric CD163 with Hb tetramer?

- In Figure 3D, the label "Hb-bound dimer" should be corrected to "Hb-bound trimer". Furthermore, while the authors compared unliganded CD163 dimer with Hb-bound CD163 trimer, it would be informative to also include a comparison with the HpHb-bound CD163 dimer solved previously by the authors.

- In Figure S1, it is suggested that the binding parameters between Hb and CD163 be shown for each individual experiment, as is currently done for HpSPHb/CD163.

Reviewer #2: In this manuscript, the authors investigate how the scavenger receptor CD163 binds and detoxifies free haemoglobin (Hb) during haptoglobin depletion. Using surface plasmon resonance and cellular uptake assays, they confirm CD163 binds and internalizes tetrameric Hb with submicromolar affinity (Kd = 382 nM), compared to its tighter binding to the HpSPHb complex (Kd = 1.18 nM). Cryo-EM analysis reveals a distinct 3:1 binding stoichiometry, demonstrating that a complete trimeric CD163 assembly is required to bind a single Hb tetramer, a finding further validated by analytical ultracentrifugation. By elucidating the structural basis for CD163's "ligand promiscuity," this study explains how the receptor adapts to clear toxic Hb and prevent oxidative damage, making it a compelling contribution for PLOS Biology readers.

While the structural insights presented are valuable, the manuscript currently lacks essential cryo-EM validation metrics. The omission of Fourier shell correlation (FSC) curves, angular distributions, and local-resolution maps precludes a thorough evaluation of the map and model quality. Additionally, several structural figures lack sufficient detail and clarity for readers to properly interpret the authors' claims. Because these are fundamental requirements for any structural biology publication, I recommend a major revision. The authors must address the following points before the manuscript can be considered suitable for publication.

Major Revisions

• The manuscript currently lacks essential cryo-EM validation data, preventing a thorough assessment of map and model quality. The authors must provide Fourier shell correlation (FSC) curves, angular distribution plots, and local-resolution maps for Map 1, the local refinement of Map 1, and Map 2. These metrics should be included as supplementary figures.

• In Lines 102-103, the authors state, “The CD163 base, CD163 arms of subunits A and B and the Hb tetramer were well resolved, while arm of subunit C was less well resolved.” Without the accompanying local-resolution maps requested above, the terms "well resolved" and "less well resolved" are subjective and difficult to interpret. Providing local-resolution maps colored directly onto the surface representation using tools like Chimera or ChimeraX will clarify these statements.

• In Lines 106-107, the authors claim that domains SRCR5-9 form a triangular base through calcium-mediated interactions, similar to the HpHb-bound structure. However, no structural illustrations are provided to support this. The authors should include a figure panel either in the main figures or supplementary figures demonstrating these specific calcium-mediated interactions within the CD163-Hb complex.

• Figure 3B is intended to demonstrate the interactions between the CD163 arms and the Hb subunits, but it falls short. Crucially, the authors rely solely on EM maps to illustrate these interfaces. Because map contours can be adjusted, this visual representation can be misleading when assessing whether two entities are genuinely interacting. Furthermore, the specific calcium-ion-mediated interaction referenced in Lines 145-146 is not visible. The authors should update Figure 3B to display the fitted atomic models within the transparent EM density, explicitly highlighting the key interacting residues and calcium ions. Example of such figures can be found in two papers reporting cryo-EM structures of CD163-HpHb complex: https://doi.org/10.1038/s41467-024-55171-4 and https://doi.org/10.1371/journal.pbio.3003264

Minor Revisions

• In Lines 131-133, the authors describe the aligned SRCR5-9 base architectures as having "an extremely similar base architecture, with an overall RMSD of 1.9 Å." An overall RMSD of 1.9 Å represents a moderate structural deviation, so the use of the word "extremely" should be tempered or removed. Additionally, the authors should elaborate on the specific base differences between the CD163-HpHb and CD163-Hb structures in the Figure 2B legend to better define what "little base movement" entails in this context.

• In Lines 181-185, the authors describe a complex mechanism of tilting and flexibility around the calcium-mediated interface joining SRCR5-7 and the interface between SRCR2 and 3. These movements are difficult to visualize from the text alone. The authors should consider providing a schematic illustration or diagram to clarify these dynamic transitions.

• The authors utilized different molar ratios of CD163 to Hb for the analytical ultracentrifugation experiments versus the cryo-EM sample preparation. Do the authors have explanations for these differing ratios?

• In Line 219 of the Methods, the authors state they mixed CD163 trimer with "Hb dimer" at a 1:1 molar ratio. Did the authors really use Hb dimer to make CD163-Hb(tetramer) complex for cryo-EM analysis? Or is it a typo?

Reviewer #3: The study by Zhou & Higgns reports the first structures of the CD163 in complex with hemoglobin. The authors use CryoEM in combination with binding and uptake assays to characterize the differences in CD163's affinity and engagement with substrates HpSBHb and Hb.

The study is straightforward, and the experiments appear well carried out. There are some modest points where the analysis can be more thorough, but I expect these to be easily addressed in a revision.

- I am a bit confused about the stoichiometry of the CD163 base. Dimers of CD163 are observed in CryoEM samples prepared without substrate or with HpSPHb. No dimer is noted here. This should be discussed, as it has significant implications for the assembly of the complexes.

- Relatedly, in the methods (line 271) it is stated that Hb binds trimeric CD163. Is this a conclusion solely based on the structure?

- It is curious that the Hb chains appear to interaction with different arms between the HpSPHb and Hb complexes. Are the Hb-SRCR domain interactions similar or different between complexes? Alternatively, are there obvious differences between the two complexes to explain this difference in engagement?

- In Figure 1B, there appears to be significant uptake of Hb in the absence of CD163. Was this background uptake corrected for in the EC50 calculation?

- In the introduction, HpSPHb is defined as describing a structural region within the larger complex between Hp and a Hb dimer. However, in the remainder of the paper this same term is used to describe a construct with a truncated Hp that has been engineered, expressed, and assembled to minimally capture this structural region. This should be clarified.

- The authors mention free Hb is normally low, but elevated in several disease states. If known, it would be very helpful to give molar concentrations of Hb in these states for comparison to measured CD163 affinities.

- Relatedly, in the introduction, circulating Hp is given in mg/ml. However, it would be helpful to give this in molar.

---

## [Editor Report · Decision Letter 2]

15 Apr 2026

Dear Dr Higgins,

Thank you for your patience while we considered your revised manuscript "Structural basis for haemoglobin scavenging by CD163 reveals mechanism of ligand promiscuity" for publication as a Short Report at PLOS Biology. This revised version of your manuscript has been evaluated by the PLOS Biology editors and the Academic Editor.

Based on our Academic Editor's assessment of your revision, I am pleased to say that we are likely to accept this manuscript for publication, provided you satisfactorily address the following data and other policy-related requests that I have provided below (A-E):

(A) We would like to suggest a very minor edit to the title, as follows, to conform to American spelling of haemoglobin (PLOS is an American-based journal). Please ensure you change both the manuscript file and the online submission system, as they need to match for final acceptance. I would also be grateful if this could be edited throughout the Abstract and manuscript text for consistency.

“Structural basis for hemoglobin scavenging by CD163 reveals mechanism of ligand promiscuity”

(B) Thank you for providing the structural data in the PDB/EMDB databases. However, we note that the data is currently on hold for release. We ask that you please make the structures publicly available at this stage before publication.

(C) Please also ensure that each of the relevant figure legends in your manuscript include information on *WHERE THE UNDERLYING DATA CAN BE FOUND*, and ensure your supplemental data file/s has a legend.

(D) Per journal policy, if you have generated any custom code during the course of this investigation, please make it available without restrictions. Please ensure that the code is sufficiently well documented and reusable, and that your Data Statement in the Editorial Manager submission system accurately describes where your code can be found. More information on our Code Policy, what and how to share can be found here: https://journals.plos.org/plosbiology/s/code-availability

(E) Please note that per journal policy, the model system/species studied should be clearly stated in the abstract of your manuscript.

We expect to receive your revised manuscript within two weeks.

*Published Peer Review History*

*Press*

Best regards,

Richard

Richard Hodge, PhD

rhodge@plos.org

PLOS

---

## [Editor Report · Decision Letter 3]

21 Apr 2026

Dear Matthew,

On behalf of my colleagues and the Academic Editor, Yan Zhang, I am pleased to say that we can accept your manuscript for publication, provided you address any remaining formatting and reporting issues. These will be detailed in an email you should receive within 2-3 business days from our colleagues in the journal operations team; no action is required from you until then. Please note that we will not be able to formally accept your manuscript and schedule it for publication until you have completed any requested changes.

PRESS

Best wishes,

Richard

Richard Hodge, PhD

rhodge@plos.org

PLOS
